# The role of social media on COVID-19 preventive behaviors worldwide, systematic review

**Eneyew Talie Fenta**[1]\*, **Eyob Ketema Bogale**[2], **Tadele Fentabel Anagaw**[2]

**1** Department of Public Health, College of Medicine and Health Sciences, Injibara University, Injibara, Ethiopia, **2** Health Promotion and Behavioral Science Department, College of Medicine and Health Science, Bahir Dar, Ethiopia

\* eneyew89@gmail.com

## Abstract

### Background

The likelihood of COVID-19 spreading from one individual to another is impacted by personal factors, preventive actions taken, and the quantity and length of exposure. Social media instantly shares health information with the public so medical professionals can interact with them.

### Methods

The review used Preferred Reporting Items for Systematic Reviews and Meta-Analysis (PRISMA) checklist. We retrieve articles by using keywords on Pub Med, Cochrane Library, and grey literatures. The Joanna Briggs Institute (JBI) Meta-Analysis of Statistics Assessment and Review Instrument was used to check the quality of articles. The main qualitative synthesis was including comments related to the characteristics of the studied populations, the study period, and the main results obtained from the social media platforms.

### Result

This study includes 32 studies conducted in 20 countries globally. Most of the studies discussed that during the pandemics different types of social medias were utilized to provide knowledge to change the attitude of the people and helps to practices COVID-19 preventive behaviors. By enabling people to seek and share knowledge, socialize, and find pleasure, social media use encourages people to adopt preventive behaviors. This review demonstrated a substantial relationship between higher levels of precautionary behaviors and factors such as educational status, knowledge, fear, and reading medical pages on social media. According to the studies, watching videos is a reliable means to get accurate information, and watching animated films posted on social media can help people learn more about COVID-19 preventive strategies.

**Data Availability Statement:** All relevant data are within the manuscript and its Supporting Information files.

**Funding:** The author(s) received no specific funding for this work.

**Competing interests:** The authors have declared that no competing interests exist.

## Conclusion

In these urgent times, social media could even help with quick information availability; misinformation or inadequate understanding can cause misunderstandings within the community. This analysis revealed that following medical pages on social media, having knowledge, feeling afraid, and having an educational background were all strongly correlated with greater levels of aware preventive behavior. Therefore, it is advised for policy makers to develop social behavioral change health communication strategies, guidelines on COVID-19 prevention behaviors. Health care providers should prepare health learning materials, and provide accurate, updated and timely information using all forms of social media to correct mis- information, misperceptions, depression, and anxiety for better control of the pandemics. Future research shall focus on evaluating effectiveness of each social media platform interventions during such emergency crisis.

## Background

After being originally identified in China in December 2019, the severe acute respiratory syndrome coronavirus 2 disease, or SARS-COV-2, has persisted as a global public health concern. The likelihood of COVID-19 spreading from one individual to another is impacted by personal factors, preventive actions taken, and the quantity and length of exposure [1–3]. When an infected individual speaks, sneezes, coughs, or comes into close contact with another sick person, COVID-19 spreads through secretions, saliva, and respiratory droplets. To avoid the spread of COVID-19, the World Health Organisation (WHO) advises adopting the following preventive measures: staying indoors, covering coughs and sneezes, wearing masks, keeping hands clean, staying a safe distance, taking antiviral medicine, and getting vaccinated. People's preventive actions during the COVID-19 pandemic determine how well COVID-19 is controlled [4–7].

There were only 20 COVID-19 deaths reported between January 3 and January 9, 2023, despite projections of 300,000 deaths in China up until April 1, 2023, and one million deaths in 2023 [8,9]. At present, the World Health Organization has recorded over 664 million confirmed cases of COVID-19 and over 6.7 million deaths worldwide as of January 24, 2023 [10]. The patients with a history of underlying diseases had a 1.41-fold increased risk of dying compared to those without a history of illness. Additionally, 20.3% of patients disregarded hygiene recommendations, including hand washing, social distancing, and wearing face masks after discharge, and had infected others in their homes [11].

Social media refers to online platforms that enable users to communicate and share material, enabling groups and individuals to interact and exchange information [12,13]. Social media is used to provide immediate health information that health care providers are able to communicate with the community. It is used to facilitate communication between healthcare providers and patients to exchange important health information which is fast, cost-effective, and two-way communication for effective preventive behavioral change [14,15].

The community received their health information primarily from social media. Social media platforms like Twitter, Facebook, YouTube, TikTok, Instagram, and LinkedIn have been utilized to spread public awareness of the COVID-19 epidemic, encourage healthy behavior, and improve community health outcomes using textual content, voice messages, and video clips [16,17]. Social media makes it possible to keep up with current events and provide

information on disease-related infection prevention guidelines. Social media health posts offer advice on how to stop the virus from spreading, including how to stay away, wear masks, and wash your hands frequently [16,18–20].

Social media can be used to create virtual communities and deliver news and updates in real time. Two categories of information on the epidemic have garnered public interest on social media. These include detailed case information, such as the movement paths of individual confirmed or suspected patients, their travel history, and the trains or flights they have taken, as well as daily statistical information, such as confirmed cases, new suspected cases, recovered cases, and deaths, both in cumulative numbers and daily updates [21]. Social media is an essential channel for health communication that the general population can use to get online information [7,22,23].

Social media plays a significant role in the rapid exchange of knowledge that leads to preventive actions being taken. In order to encourage users' preventive actions against behavioral illnesses, social media are useful instruments for health communication. People were said to have been exposed to more COVID-19 information that encourages preventive behaviors when they used social media more frequently and for longer periods of time [20,24–26]. Early in the outbreak, the researchers noted that misinformation about lockdowns and impending government rules was conveyed via text messages. Misinformation regarding misconceptions about the disease is still a concern when it comes to COVID-19 pandemic prevention, despite the efforts of most social media sites to combat it. Effective risk communication for unknown infection threats depends on trust, and behavioral change is influenced by confidence in perceived information [1,27–31]. The vast quantity of COVID-19 information made possible by social media was linked to the public's perception of COVID-19 susceptibility and severity with increased COVID-19 preventive behaviors. The participants who were exposed to COVID-19 related information on social medias had a positive effect on preventive behaviors [32–34].

During the COVID-19 pandemic, a wider range of information can be shared on social media because to people's anxiety and willingness to share information, even if they are unsure of it at first [35,36]. The rapid spread of pandemic information on social media has likely raised people's awareness of the symptoms of COVID-19, the means by which the virus spreads, and the need for preventive measures. However, social media is also rife with false information, rumors, and misconceptions about the illness, which may have contributed to an increase in people's anxiety, worry, and fear [37,38]. According to the studies, lowering the quantity of false information could reduce the severity of disease. Reducing the transmission of false information or the inclination to believe it could lower the prevalence of infectious diseases. For all diseases, it was anticipated that stage 2 conditions would return to stage 1 after roughly 20% of the population was immunized against false information [39,40]. As a result, a lot of public health organizations disseminated customized information to the public via social media platforms in order to encourage preventative measures and debunk common misconceptions [18,40–43]. Therefore, this systematic review aimed to show the role of social media on COVID-19 preventive behaviors globally.

## Methodology

**Objective.**  The primary objective of this review is to synthesize evidence on the role of social media on COVID-19 preventive behaviors globally.

*Review question*. What is the role of social media platforms on COVID-19 preventive behaviors?

**Searches.**  This systematic review and meta-analysis were done using Preferred Reporting Items for Systematic Reviews and Meta-Analyses (PRISMA) guidelines. A search strategy was

implemented on PubMed, Cochrane Library, and grey literatures, registers, websites, organizations, reference lists and other sources searched or consulted to identify studies, which were systematically searched online to retrieve related articles until June 30/2023 using keywords. ((((((((((((social media) OR (Instagram)) OR (YouTube)) OR (Facebook)) OR (Twitter)) OR (TikTok)) OR (LinkedIn)) AND (COVID-19 preventive behavior)) OR (sars-cov-2 preventive behavior)) OR (Coronavirus disease preventive behavior)). We also manually scanned the references of all articles in which full-text reading was performed not to miss additional articles. The review protocol is available on PROSPERO (ID: CRD42023450425).

**Eligibility criteria.** For this review PICO mnemonic (Population, Intervention, control and outcome) approach was used. Participants/population, adult population who uses any form of social media. Outcome: The aim of this study is to synthesize the role of social media on COVID-19 preventive behaviors globally. social media is an important and popular way to disseminate messages to a wide audience, especially those who are frequent users of social media. Intervention, social media are web-based resources or programme that let individuals or communities communicate by sending and receiving text messages, photos, and other types of content. Social networking sites were a part of social media. (e.g., Facebook, YouTube and Instagram), chat or message. Comparator(s)/control, People who adhere COVID-19 preventive behaviors without using social media. This systematic review includes all articles that meet the following criteria: articles that examine the role of social media and COVID-19 preventive behaviors were included. Studies without full text or protocols, conference abstract, books, if they lacked sufficient information on important details of the intervention, articles written in languages other than English, short reports, letters to the editor, and discussions were excluded from the present systematic review.

*Data extraction.* Articles extracted from search engines were exported to a Microsoft Excel spreadsheet after the removal of duplicates. Studies retrieved by using search terms from all databases and additional sources were screened for inclusion criteria. Then, articles that fulfilled the inclusion criteria were undertaken full-text review for admissibility and extraction. The preferred Reporting Item for Systematic Review and Meta-analysis (PRISMA) flowchart was used throughout all steps.

For each included article, the following information was extracted: author, publication year, country, sample size, sampling method, location of data collection, study design, PICOS, follow-up period, citations for each study, measures of COVID-19 preventive behaviors, the role of social media and COVD-19 preventive behaviors and factors associated with social media utilization and COVD-19 preventive behaviors. Titles and abstracts of the studies retrieved were screened independently by two review authors to identify all the articles that potentially meet the inclusion criteria. The full text of the eligible studies was retrieved and independently assessed for eligibility by two review team members. Two researchers performed the data extraction. Disagreements were resolved in consensus or by a third reviewer.

*Quality assessment.* We have used 13-point JBI Critical Appraisal Checklist for Randomized Controlled Trials, for five RCT studies and 8-point JBI Critical Appraisal Checklist for Analytical Cross-Sectional Studies for 27 studies. Using the tool as a protocol, the reviewers (ETF, TDA) used the blinded review approach to evaluate the quality of the original articles. Those studies, scores above the mean for both cross-sectional and interventional studies were considered to have good quality and included in the review. Discrepancies in the quality assessment was resolved through other authors (EKB).

*Strategy for data synthesis.* The extracted data was presented in tables and classified into main topics presenting the main outcomes identified throughout the systematic review process. The main qualitative synthesis was including comments related to the characteristics of the studied populations, the study period, sample size, study design, the role of each social

media platforms on COVID 19 preventive behavior adoption. It was impossible to perform a meta-analysis or best evidence narrative synthesis due to the high heterogeneity in the study population, social media platforms, and measuring instruments utilized on study participants to assess social media and COVID-19 preventive behaviors.

## Result

In this systematic review, one thousand six hundred fifty-eight (1658) articles were retrieved from all searched database sources. After the removal of 1375 duplicates, 283 articles were assessed using title and abstract then, 196 articles were excluded since not report the relationship between social media and COVID-19 preventive behaviors. Fifty-four (87) articles were included as shown in Fig 1 in the full-text review,55 articles exclude, and then 32 articles were remained for final systematic review as shown in Fig 1.

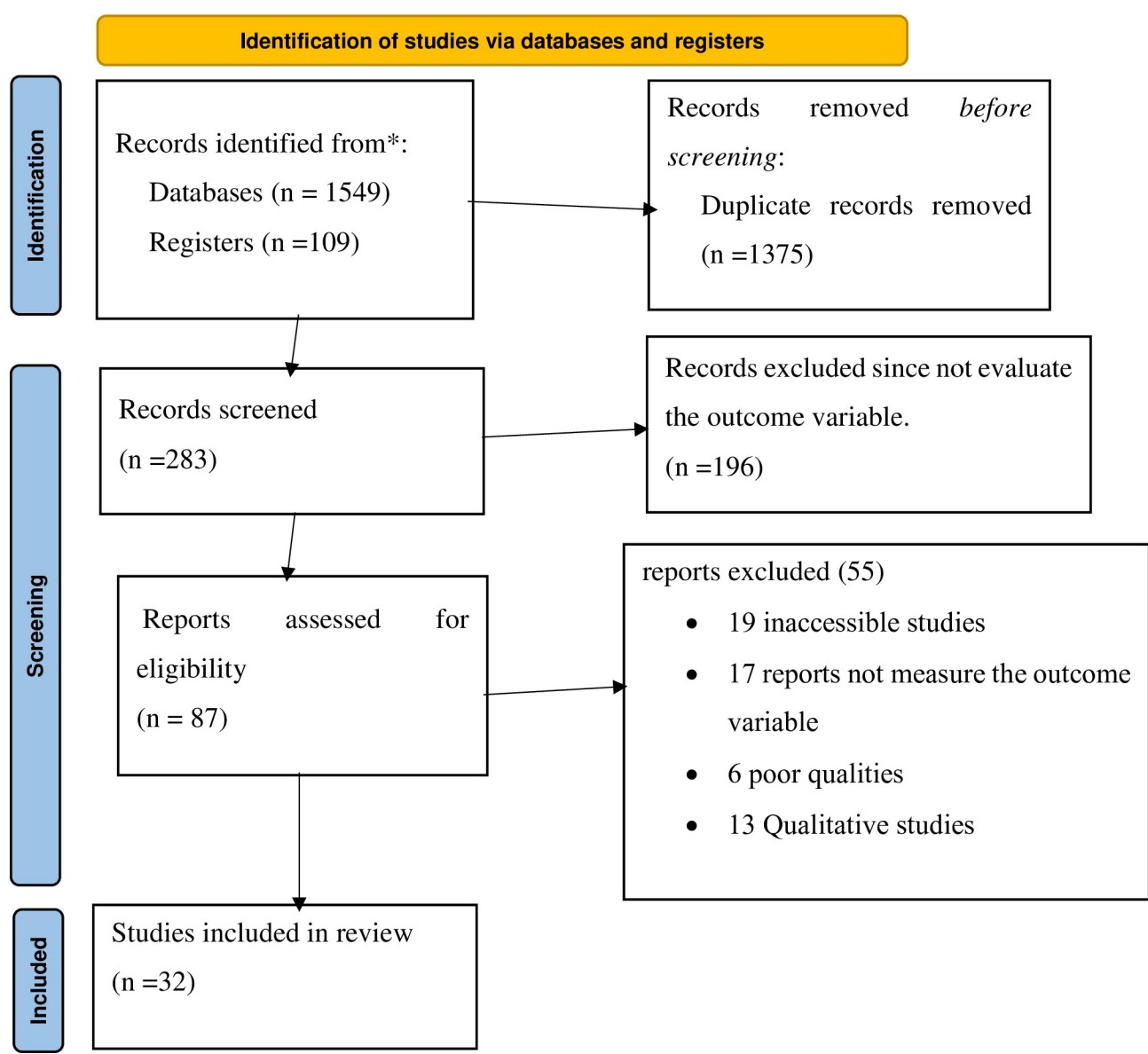

**Fig 1. PRISMA flow diagram of study selection for the role of social media on COVID-19 preventive behaviors worldwide, systematic review.**

## Study characteristics

This study includes 32 studies conducted in 20 countries globally in which, six studies in USA [44–49], China five [34,50–53] Malaysia three,[54–56], Korea two[57,58] Pakistan [59], Spain [60], Poland and Jordan [61], Indonesia [62], Guatemala [63], Japan [64], Jordan [31], Vietnam [65], Bangladesh [66], Ethiopia [67], Italy [68], Nigeria [69], Saudi Arabia [70], Canada [71], SSA [72], and in United States, Mexico, the United Kingdom, Germany, and Spain, [73] one study in each country. Most of the studies were cross-sectional study design except five, which were experimental studies. As explained in Table 1, the total sample size was 37,085 with maximum sample 14,482 sample in USA among experimental studies and minimum of 100 sample size in USA with cross-sectional study (Table 1).

According to a study conducted in Bangladesh, using creative social media to read essays, write articles, viewing flyers and photos, and watch movies increases awareness of COVID-19 preventive actions. These materials can be shared on social media sites like Facebook, WhatsApp, and YouTube. Studies conducted in China found a substantial correlation between trust in doctors' social media accounts and preventive behaviors related to the coronavirus. Individuals who place greater trust in unofficial information obtained from social media and interpersonal interactions are more likely to wear masks and practice hand washing as preventive measures [34,50–53,66].

Students in Indonesia utilize social media extensively, which has the effect of raising awareness about the prevention of Covid-19 transmission by mask wear, physical distance, and soap washing. In languageless Guatemalan Exposure to graphic interchange formats was statistically linked to improved intentions, self-efficacy, and outcome expectancy beliefs regarding physical distancing; handwashing intentions and outcome expectancy beliefs; and mask-wearing intentions and self-efficacy. The studies discussed individuals who used social media effectively to obtain information about COVID-19 were 9.5 times more likely than non-users to engage in COVID-19 preventive measures [62,63,67].

According to a Korean study, people are more likely to consider preventive behavior if they utilize social media more regularly for news and information regarding COVID-19. When it came to searching for information on COVID-19 pandemic prevention, social media in Japan was more predictive than mass media. Furthermore, following medical pages on social media, feeling afraid, and having knowledge were all strongly linked to higher levels of preventive behavior in Jordan. One particularly effective way to use social media platforms to promote dialogue is by raising public awareness and sensitization about the COVID-19 prevention campaign. Facebook campaigns, for example, have shared information and sparked discussions about pertinent COVID-19 emergency topics between scientists and the general public [57,58,64,68].

Positive correlations were shown between public health awareness, COVID-19 preventative behaviors, and the elements of a social media campaign in studies conducted in Poland and Jordan. Social media is used by 93.2% of participants in Saudi Arabia to obtain information on COVID-19, but it has no discernible effect on the adoption of preventive measures. Social media usage and COVID-19 misperceptions appear to be associated in Canada, indicating that misinformation on social media plays a role in contributing to these misperceptions. People who viewed face masks as useful in avoiding COVID-19 were more likely to utilize social media as their primary source of COVID-19 information, according to a study conducted in six sub-Saharan African nations [61,70,71]. According to research conducted in the United States, TikTok's general handwashing videos are crucial for educating the public about proper handwashing practices. Emergency physicians are seen to be more successful in disseminating COVID-19 health recommendations when they share their personal stories on Twitter.

**Table 1. The role of social media on COVID-19 preventive behaviors worldwide, systematic review.**

| First author/ Pub year | Study design | Country | Sample size | Social media and COVID-19 preventive behaviors | Quality score |
|---|---|---|---|---|---|
| Hernández-García I et al. [60]/2020 | cross-sectional study | Spain | 129 | videos to be viewed in order to obtain reliable information on hand washing practice. | 7 |
| Basch CH, et al. [44]/ 2022 | cross-sectional study | USA | 100 | Wash your hands accounted for 93.3% of the total views. Coverage of the important steps involved in handwashing, such as drying hands, was minimal as was relevant background information. | 6 |
| Abuhashesh MY etal. [61]/2021 | cross-sectional study | Poland and Jordan | 1149 | positive relationships between the components of a social media campaign, public health awareness, and behavioral change during COVID-19. | 8 |
| Smail EJ et al.[46]/ 2023 | cross-sectional study | USA | 1057 | Increasing social media use was associated with engagement in more COVID-19 precautionary behaviors. | 7 |
| Zhang SX,et al [56]/ 2020 | cross-sectional study | Malaysia | 674 | More time spent on social media was positively associated with hand washing for males with three or more children. However, for males without children, social media use was negatively associated with 5handwashing. | 7 |
| Wardani EM, etal/ 2020[62] | Experimental | Indonesia | 5400 | Social media is effectively used to provide health education because it can increase student knowledge and influence behavior in preventing covid-19 transmission. | 9 |
| O'Brien N, et al/2022 [63] | Experimental | Guatemala | 308 | GIF exposure significantly improved participants' self-efficacy, intention and belief on hand washing behavior. | 8 |
| Yu J, et al[53] /2022 | cross-sectional study | China | 122 | People who trust in informal information from social media and interpersonal communication would be more likely to adopt mask wearing and hand washing protective behaviors. | 5 |
| Suzuki T, et al [64] (2021) | cross-sectional study | Japan | 987 | social media had stronger predictive power to change attitude through time. | 6 |
| Breza E, et al. [45]/ 2021 | Experimental | USA | 410 | Social media messages recorded by health professionals before the winter holidays in the United States led to a significant reduction in holiday travel and subsequent COVID-19 infections. | 6 |
| Yassin A,et al [31] (2022) | cross-sectional study | Jordan | 827 | The variables that were significantly associated with higher levels of precautionary behaviors were knowledge, feeling of fear, and following medical pages on social media. | 8 |
| Niu Z, et al/2021[51] | cross-sectional study | China | 464 | The results suggested that the vulnerable populations' engagement in coronavirus-related preventive behaviors were significantly associated with barriers, benefits, self-efficacy, trust in doctors' social media, and trust in TV for COVID-19-related information. Besides, barriers, benefits, self-efficacy, trust in doctors' social media, and trust in TV mediated the effects of health literacy on preventive behaviors. | 7 |
| Duong HT, et al [65]/ 2021 | cross-sectional study | Vietnam | 360 | Results indicated that interpersonal communication mediated the effect of social media campaign exposure on intentions to stay at home. | 7 |
| Tsoy D,et al [49]/2022 | cross-sectional study | USA | 306 | social media would have a positive impact on staying at home intentions. But if users instead perceived messages as being fabricated, amplified, or false, the alert would stay dangerously low. | 8 |
| Islam MM,et al [66] /2021 | cross-sectional study | Bangladesh | 265 | Creative social media increase knowledge on COVID-19 prevention measures by using seven online activities which are awareness of Covid-19 spread, wearing the mask, hand washing, precaution before touching the face, nose and eyes, avoiding gathering, social distances, and knowing Covid-19 symptoms. These activities are exhibited in reading essays/writings, seeing photos and flyers, watching videos posting writings, pictures, flyers and videos on social media platforms like Facebook, WhatsApp and YouTube. | 5 |
| Wang H,et al[48]/2021 | cross-sectional study | USA | 500 | The excessive information disseminated on social media platforms and other sources is closely related to the dynamics of the general public's health beliefs. | 6 |
| Wu G, et al [52] /2020 | cross-sectional study | China | 592 | Social media information sources, unofficial social media caused both Wuhan and non-Wuhan urban citizens to have higher levels of panic than official media but had no significant impacts on their preventive behaviors. | 7 |
| Lu J,et al [34]/2023 | cross-sectional study | China | 739 | Social media empowers individuals in terms of knowledge seeking, knowledge sharing, socializing and entertainment to promote preventive behaviors at the individual level by increasing each person's perception of collective efficacy and social cohesion. | 7 |

(*Continued*)

**Table 1.** (Continued)

| First author/ Pub year | Study design | Country | Sample size | Social media and COVID-19 preventive behaviors | Quality score |
|---|---|---|---|---|---|
| Sadore AA, et al [67]/ 2021 | cross-sectional study | Ethiopia | 372 | Study participants who had good use of social media to get COVID-19–related information were 9.5 times more engaged in COVID-19 preventive practices than non-users. The study participants who had a high-risk perception of COVID-19 were 2.6 times more engaged in COVID-19 practices compared with study participants who had a low-risk perception of COVID-19. | 6 |
| Graffigna G, et al [68]/ 2020 | cross-sectional study | Italy | 976 | 10% of the audience showed an active engagement with the campaign, by expressing likes, by writing comments on, or sharing its contents on Facebook and LinkedIn. Facebook generally appears a more suitable platform for engaging with the audience and as a means to convey public health information in a lively manner. | 8 |
| Choi D-H, et al, /2023 [57] | cross-sectional study | Korea | 1500 | Social media use is positively associated with social norms, which may have a positive relationship with COVID-19 preventive behavior. | 6 |
| Mohammed F,[55]/ 2023 | cross-sectional study | Malaysia | 488 | The findings showed that perceived risk, e-health literacy, public awareness, and health experts' participation influence public protective behavior when using social media to share COVID-19-relevant content. | 5 |
| Okpara CV,et al [69]/ 2021 | cross-sectional study | Nigeria | 470 | The result showed that recall of messages theme in COVID-19 YouTube animated cartoons significantly predicts health behavior of social media users. | 6 |
| Yu S,et al [59] /2022 | cross-sectional study | Pakistan | 348 | The findings approve that attitudes toward social media use in the pandemic have positively mediated the relation between distancing and practices for social media use amid the crisis of COVID-19. | 5 |
| Solnick RE, et al [47]/ 2021 | experimental study | USA | 2007 | The public health messages delivered by physicians and personal messages elicited stronger emotions, greater changes in attitudes, and an increased willingness to disseminate the message than when federal officials delivered impersonal messages. | 6 |
| Alrasheed M,et al [70]/ 2022 | cross-sectional study | Saudi Arabia | 1500 | 93.2% of participants use social media for COVID-19 related information. High social media exposure was significantly associated with higher risks of anxiety, depression, and higher levels of COVID-19 risk perception. However, social media has no significant impact on the adoption of preventive behavior. | 7 |
| Mat Dawi N,et al /2021[54] | cross-sectional study | Malaysia | 404 | perception of e-government information and services and perception of social media were found to be significant predictors of attitude toward preventive behavior. | 6 |
| Bridgman A,et al [71]/ 2020 | cross-sectional study | Canada | 500 | Exposure to social media is associated with misperceptions regarding basic facts about COVID-19 while the inverse is true for news media. These misperceptions are in turn associated with lower compliance with social distancing measures. | 5 |
| Lee J,et al/2021[58] | cross-sectional study | South Korea | 1000 | The results reveal that the perceived characteristics of online news and social media influence preventive actions through the trust in citizens or in government. | 7 |
| Iyamu I,[72] /2021 | cross-sectional study | SSA | 1988 | The respondents who used social media were more likely to agree that face masks were effective compared with those who did not. | 7 |
| Liu PLJSS,et al [50]/ 2021 | cross-sectional study | China | 511 | Results indicated that personal responsibility partially mediated the relationship between COVID-19 information consumption on social media and preventive behaviors. | 6 |
| Vandormael A, et al [73]/2021 | RCT | USA, & Europe | 14,482 | Short, wordless, animated videos, distributed by health authorities via social media, may be an effective pathway for rapid global health communication during health crises. | 10 |

Intentions to stay at home were influenced by social media campaign exposure, although in Vietnam, this effect was mitigated by interpersonal communication [44–49,65].

Perceived risk, e-health literacy, public awareness, and the involvement of health experts all influence COVID-19 public protective behavior through social media, according to study results from Malaysia. According to a Pakistani study, social media plays a significant impact in social alienation and COVID-19 knowledge. A study conducted in Spain revealed that watching movies is a dependable way to learn about the proper way to wash your hands. In research found in Nigeria, among social media users who watch the COVID-19 YouTube animated cartoon, self-efficacy, task self-efficacy, coping self-efficacy, and outcome expectancy significantly predict health behavior sustainability. According to a study conducted in the US,

Mexico, the UK, Germany, and Spain, quick, wordless animated films that are shared on social media by health authorities could be a useful tool for quick worldwide health communication in times of emergency [54–56,59,60,69,73].

## Discussion

In this systematic review 32 articles were included on the role of social media on COVID-19 preventive behaviors globally. Most of the studies discussed that during the pandemics different types of social medias were utilized to provide knowledge to change the attitude of the people and helps to practices COVID-19 preventive behaviors. The use of social media empowers individuals in terms of knowledge seeking, knowledge sharing, socializing and entertainment to promote preventive behaviors. Social media had stronger predictive power to browse information on COVID-19 pandemic prevention [34,46,56,57,61,62,64,67]. This was consistent with the studies discussed that social media platforms played a vital part in controlling pandemics and were useful communication tools for supporting public health initiatives and information dissemination. During the COVID-19 pandemic, social media campaigns were helpful in spreading the word about personal hygiene and social distancing. They also play a significant role in assisting with preparation, reaction, and recovery during medical emergencies [28,30,74–77].

Prior to the US winter holidays, health professionals recorded social media postings that significantly decreased holiday travel and the ensuing COVID-19 infections. Public opinion on health on social media may be somewhat influenced by the dynamics of the pandemic, news, scientific and nonscientific events, and even the relevant tweets that have previously been posted. It is believed that medical professionals who share personal stories on Twitter are more successful at disseminating COVID-19 health recommendations. There was a substantial correlation found between trust in doctors' social media and preventive behaviors connected to the coronavirus. Medical professionals' personal and public health messages evoked more powerful feelings, more significant attitude shifts, and more readiness to spread the word than did government authorities' impersonal remarks [45,47,51]. This was similar to a study that revealed numerous healthcare organizations, physicians, and hospitals have set up Facebook, Twitter, and YouTube accounts to reach out to their patients. Additionally, medical personnel used social media to avoid outbreaks and provide health advice. An increasing number of doctors, scientists, and other health professionals are using Twitter to effectively disseminate information during emergencies and during public health crises in order to communicate suggestions[78–80].

According to this study, social media use was positively correlated with misperceptions about COVID-19, indicating that disinformation on social media may have contributed to some of these misperceptions. Even social media would help people intend to stay at home; if users thought that posts were fake, exaggerated, or manufactured, the alarm level would remain dangerously low [48–50,71]. This was in line with research showing the value of social media in reducing false information, raising public awareness of reliable health-related information, and improving real-time surveillance during pandemics. Even though quick access to information is crucial in these emergency situations, misunderstandings within the community can result from poor comprehension or from erroneous or incorrect information. The COVID-19 pandemic has given rise to more false information on social media, including news that presents two opposing perspectives of the virus. This misinformation affects public safety and makes crisis management more difficult, necessitating health care providers to actively combat both the virus and its associated misinformation at the same time[81–84].

This review demonstrated a substantial correlation between higher levels of precautionary behaviors and educational status, knowledge, fear, and following medical pages on social media. The study also revealed an important connection between the vulnerable populations' participation in coronavirus-related preventive behaviors and the hurdles, benefits, self-efficacy, and confidence that doctors' social media accounts provide regarding COVID-19-related information. When compared to study participants who saw COVID-19 as low-risk, those who perceived COVID-19 as high-risk were more involved in COVID-19 behaviors [31,34,44,51,55,62,67]. This study conducted in Uganda and China revealed that individuals with good knowledge engaged in COVID-19 preventative behavior [76,85,86].

There was a strong correlation found between high social media exposure and increased levels of COVID-19 risk perception, anxiety, and depression. Significant determinants of attitude towards preventative behavior were found to be perceptions of social media and e-government information and services. In turn, these misconceptions are linked to a decrease in adherence to social distancing protocols [54,67,70,71]. This has similar finding with the study discussed that dissemination of misinformation about health can lead to unnecessary and undesirable outcomes such as fear, anxiety, misunderstanding of the disease, and problems in the patient-doctor relationship. The studies also reported that social media news may increase anxiety about dangers of the virus, subsequently leading to uptake of precautionary behaviors [19,79,84,87,88].

According to the studies, watching movies is a good way to learn about proper hand washing techniques. Through reading, viewing, and watching films, as well as publishing writings, images, flyers, and videos on social media sites like Facebook, WhatsApp, and YouTube, creative social media raises awareness of COVID-19 preventative methods. During health crises, the animated movies that health officials share on social media could be a useful tool for quick global health communication. The findings demonstrated that social media users' COVID-19 preventative behavior is significantly predicted by their recollection of the themes from COVID-19 animated cartoons on YouTube [60,63,66,68,69,73]. This was comparable with the study findings reported that videos are attractive and were previously proven to be very effective for health education and behavioral change[15,59,89,90].

## Strengths and limitations

The strength was doing this global systematic review on social media role on COVID 19 preventive behaviors which is the global public health emergency for better utilization of social media and controlling COVID19.its limitation was due to heterogeneity of the study we could not done meta-analysis.

## Conclusion

In this systematic review most of the studies discussed that during the pandemics different types of social medias were utilized to provide knowledge to change the attitude of the people and helps to practices COVID-19 preventive behaviors. Social media helps people to seek and share knowledge, connect with others, and find enjoyment and amusement to support preventive behaviors. When searching for information on COVID-19 pandemic prevention, social media exhibited a better predictive capacity. In these urgent times, social media could even help with quick information availability; misinformation or inadequate understanding can cause misunderstandings within the community.

According to the studies, watching videos is a dependable way to get accurate information, and watching animated films posted on social media can help people learn more about COVID-19 preventive strategies. There was a strong correlation found between high social

media exposure and increased levels of COVID-19 risk perception, anxiety, and depression. In turn, these misconceptions are linked to a decrease in adherence to social distancing protocols. This review demonstrated a substantial relationship between higher levels of precautionary behaviors and factors such as educational status, knowledge, fear, and following medical pages on social media. Therefore, it is advised for policy makers to develop social behavioral change health communication strategies, guidelines on COVID-19 prevention behaviors. Health care providers should prepare health learning materials, and provide accurate, updated and timely information using all forms of social media to correct mis- information, misperceptions, depression, and anxiety for better control of the pandemics. Future research shall focus on evaluating effectiveness of each social media platform interventions during such emergency crisis.

## Supporting information

**S1 Fig. PRISMA flow diagram of study selection ON The role of social media on COVID-19 preventive behaviors worldwide, systematic review.**
(DOCX)

**S1 Table. The role of social media on COVID-19 preventive behaviors worldwide, systematic review.**
(PDF)

## Author Contributions

**Conceptualization:** Eneyew Talie Fenta, Tadele Fentabel Anagaw.

**Formal analysis:** Eneyew Talie Fenta, Eyob Ketema Bogale, Tadele Fentabel Anagaw.

**Investigation:** Eneyew Talie Fenta.

**Methodology:** Eneyew Talie Fenta, Eyob Ketema Bogale.

**Software:** Tadele Fentabel Anagaw.

**Supervision:** Tadele Fentabel Anagaw.

**Validation:** Eyob Ketema Bogale, Tadele Fentabel Anagaw.

**Writing – original draft:** Eneyew Talie Fenta.

**Writing – review & editing:** Tadele Fentabel Anagaw.

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
