## [Decision Letter · Decision Letter 0]

13 May 2024

PONE-D-24-11425The role of social media on COVID-19 preventive behaviors worldwide, systematic reviewPLOS ONE

Dear Dr. Talie Fenta,

Thank you for submitting your manuscript to PLOS ONE. After careful consideration, we feel that it has merit but does not fully meet PLOS ONE’s publication criteria as it currently stands. Therefore, we invite you to submit a revised version of the manuscript that addresses the points raised during the review process.

We look forward to receiving your revised manuscript.

Kind regards,

Mukhtiar Baig, Ph.D.

Academic Editor

PLOS ONE

Journal Requirements:

2. Please amend either the abstract on the online submission form (via Edit Submission) or the abstract in the manuscript so that they are identical.

Reviewers' comments:

Reviewer's Responses to Questions

**Comments to the Author**

1. Is the manuscript technically sound, and do the data support the conclusions?

Reviewer #1: Yes

Reviewer #2: Yes

Reviewer #3: No

2. Has the statistical analysis been performed appropriately and rigorously? 

Reviewer #1: N/A

Reviewer #2: Yes

Reviewer #3: N/A

3. Have the authors made all data underlying the findings in their manuscript fully available?

Reviewer #1: Yes

Reviewer #2: Yes

Reviewer #3: No

4. Is the manuscript presented in an intelligible fashion and written in standard English?

Reviewer #1: Yes

Reviewer #2: Yes

Reviewer #3: No

5. Review Comments to the Author

Reviewer #1: Great job to discuss this important topic.

Joanna Briggs Institute (JBI) critical appraisal check list has different type of critical appraisal depending on the study design? Which ones were used in this study? knowing that some articles included was interventional studies.

Analysis should be splitted into interventional study --- and try to do meta-analysis----

and observational study --- you can do meta-analysis if you have clear definition of outcome variable measurement?

This is a systematic review without meta-analysis, it only includes qualitative synthesis of data. Did you design it from the beginning to be just systematic review? or you found it wasn't applicable to do meta-analysis after risk of bia assessment?

Social media has opposites role during COVID-19 pandemic, one that is positive and aid in preventive behaviour adoption, and the other is spreading mis-information. i wish the author did extract both information from articles. like how they find people didn't use social media in article samples?

Reviewer #2: 1- Systematic review registration number is not provided.

2- The rationale for the review is not described in the context of what is already known.

3- The process for selecting studies (i.e., screening, eligibility, included in systematic review, and, if applicable, included in the meta-analysis) is not stated

4- Method of data extraction from reports (e.g., piloted forms, independently, in duplicate) and any processes for obtaining and confirming data from investigators are not described.

5- Numbers of studies screened, assessed for eligibility, and included in the review, with reasons for exclusions at each stage are not given

6- Characteristics for which data were extracted (e.g., study size, PICOS, follow-up period) and citations for each study are not provided

7- Discussion: The main findings including the strength of evidence for each main outcome are not summarized considering their relevance to key groups (e.g., healthcare providers, users, and policy makers)

8- A general interpretation of the results in the context of other evidence, and implications for future research are not provided

9- I suggest to use epidemiological data from following article in the introduction:

I- Bagi HM, Soleimanpour M, Abdollahi F, Soleimanpour H. Evaluation of clinical outcomes of patients with mild symptoms of coronavirus disease 2019 (COVID-19) discharged from the emergency department. PLOS ONE. 2021;16(10):e0258697.

Reviewer #3: Comments & suggestions

Good effort by the investigators. However, there are some major concerns with the methods and analyses that need to be addressed.

Rationale & objective

Describe the rationale for the review in the context of existing knowledge.

Provide an explicit statement of the objective(s) or question(s) the review addresses.

Review question The primary objective of this review is to synthesize evidence on the role of social media on COVID-19 preventive behaviors globally

Under heading of review question, objective is written

What is the review question ??

Methodology

Following points were not dicussed adequately ‘

inclusion and exclusion criteria for the review and how studies were grouped for the syntheses is not mentioned explicitly

only PubMed, Cochrane Library, and grey literatures were accessed , specify all databases, registers, websites, organisations, reference lists and other sources searched or consulted to identify studies.

Full search strategy with search string using Boolean operators in not mentioned . Present the full search strategies for all databases, registers and websites, including any filters and limits used.

Specify the methods used to decide whether a study met the inclusion criteria of the review, including how many reviewers screened each record and each report retrieved, whether they worked independently, and if applicable, details of automation tools used in the process.

Specify the methods used to collation of data ,how consensus was developed , is there any objective criteria , how many reviewers were involved , whether they worked independently, any processes for obtaining or confirming data from study investigators, , details of automation tools if used in the process.

List and define all outcomes for which data were sought. Specify whether all results that were compatible with each outcome domain in each study were sought (e.g. for all measures, time points, analyses), and if not, the methods used to decide which results to collect.

List and define all other variables for which data were sought (e.g. participant and intervention characteristics, funding sources). Describe any assumptions made about any missing or unclear information.

You have mentioned ;

a. Outcome (In outcome aim is mentioned instead of specifying the outcome as one of the domain of PICO ) The aim of this study is to synthesize the role of social media on COVID-19 preventive behaviors globally. social media is an important and popular way to disseminate messages to a wide audience, especially those who are frequent users of social media

b. Comparator(s)/control, People who adhere COVID-19 preventive behaviors without using social media…was the comparison group there, comparison done in this study ?????

Specify the methods used to assess risk of bias in the included studies, including details of the tool(s) used, how many reviewers assessed each study and whether they worked independently, and if applicable, details of automation tools used in the process.

Quality assessment The Joanna Briggs Institute (JBI) critical appraisal check list was used ( which one ???, author haven’t specified which tool was used?

It is mentioned “Most of the studies were cross-sectional study design except five, which were experimental studies” which tool was use to assess the quality of cross sectional study and which for experimental ???

Specify for each outcome the how qualitative synthesis was done also describe the processes used to decide which studies were eligible for each synthesis and methods required to prepare the data for presentation or synthesis, such as handling of missing summary statistics, or data conversions.

RESULT

Better to use new version of the PRISMA flow diagram(PRISMA 2020)

Present assessments of risk of bias for each included study.

Its mentioned “Disagreements were resolved in consensus or by a third reviewer”. How these were resolved. no detailed info how consensus was reached . For each synthesis, briefly summarise the characteristics and risk of bias among contributing studies.

DISCUSSION

Provide a general interpretation of the results in the context of other evidence.

Discuss any limitations of the evidence included in the review.

Discuss any limitations of the review processes used explicity .

Discuss implications of the results for practice, policy, and future research.

OTHER INFO

Provide correct registration information for the review, including register name and registration number, or state that the review was not registered.( Registration number provided is CRD42023450425 but its not accessible , showing error)

Indicate where the review protocol can be accessed, or state that a protocol was not prepared.

Describe and explain any amendments to information provided at registration or in the protocol.

The report which of the following are publicly available and where they can be found data extracted from included studies; serach strategy, can it be re-run, data used for all analyses;; any other materials used in the review.

Please proofread and correct grammatical errors throughout the paper.

According to the study, lowering the quantity of false information could lessen the severity of disease outbreaks by 10% to 20% ( very strong statement without reference and grammatically incorrect)

Two categories of information on the epidemic have garnered public interest on social media. These include detailed case information, such as the movement paths of individual confirmed or suspected patients, their travel history, and the trains or flights they have taken, as well as daily statistical information, such as confirmed cases, new suspected cases, recovered cases, and deaths, both in cumulative numbers and daily updates (this para is also not supported with ref , which sites provided this info).

According to a study conducted in Bangladesh, using creative social media to read essays, write articles, view fliers and photos,….. fliers or flyer???

6. PLOS authors have the option to publish the peer review history of their article (what does this mean?). If published, this will include your full peer review and any attached files.

Reviewer #1: No

Reviewer #2: No

Reviewer #3: **Yes: **Khola Noreen

---

## [Decision Letter · Decision Letter 1]

15 Jun 2024

The role of social media on COVID-19 preventive behaviors worldwide, systematic review

PONE-D-24-11425R1

Dear Dr. Fenta,

We’re pleased to inform you that your manuscript has been judged scientifically suitable for publication and will be formally accepted for publication once it meets all outstanding technical requirements.

Kind regards,

Mukhtiar Baig, Ph.D.

Academic Editor

PLOS ONE

Reviewers' comments:

Reviewer's Responses to Questions

**Comments to the Author**

1. If the authors have adequately addressed your comments raised in a previous round of review and you feel that this manuscript is now acceptable for publication, you may indicate that here to bypass the “Comments to the Author” section, enter your conflict of interest statement in the “Confidential to Editor” section, and submit your "Accept" recommendation.

Reviewer #2: All comments have been addressed

2. Is the manuscript technically sound, and do the data support the conclusions?

Reviewer #2: Yes

3. Has the statistical analysis been performed appropriately and rigorously? 

Reviewer #2: Yes

4. Have the authors made all data underlying the findings in their manuscript fully available?

Reviewer #2: Yes

5. Is the manuscript presented in an intelligible fashion and written in standard English?

Reviewer #2: Yes

6. Review Comments to the Author

Reviewer #2: The article has been revised in the best manner and it is suitable for publication in the journal of plos one

7. PLOS authors have the option to publish the peer review history of their article (what does this mean?). If published, this will include your full peer review and any attached files.

Reviewer #2: No

---

## [Editor Report · Acceptance letter]

1 Jul 2024

PONE-D-24-11425R1 

PLOS ONE

Dear Dr. Talie Fenta, 

I'm pleased to inform you that your manuscript has been deemed suitable for publication in PLOS ONE. Congratulations! Your manuscript is now being handed over to our production team.

Kind regards, 

on behalf of

Professor Mukhtiar Baig 

Academic Editor

PLOS ONE